# PERSPECTIVE

## Dopamine inhibits gill O$_2$ chemoreception: a fishy tale excites

Colin A. Nurse 🔘

*Department of Biology, McMaster University, Hamilton, Ontario, Canada*

Email: nursec@mcmaster.ca

Handling Editors: Harold Schultz & Andrew Holmes

The peer review history is available in the Supporting Information section of this article (https://doi.org/10.1113/JP288732#support-information-section).

In air-breathing vertebrates dopamine (DA) is a well-established neurotransmitter that plays a critical role in respiratory control during normal development and acclimatization to low oxygen (hypoxia). In most species its role is inhibitory and its actions are mediated by dopaminergic D2 receptors (D2R). For example in the mammalian carotid body (CB), a major peripheral O$_2$-sensing organ, DA, released from chemoreceptor cells during hypoxia, acts in an autocrine-paracrine manner to cause pre- and postsynaptic inhibition, thereby modulating CB function together with other neurotransmitters (Leonard et al., 2018). However little is known about the evolutionary role of DA in respiratory control and whether similar pathways exist in water-breathing vertebrates. In this issue of *Journal of Physiology*, Reed and Jonz (2025) address this void in an elegant study, using a transgenic zebrafish (*Danio rerio*) line, Tg(*elavl3*:GCaMP6s), expressing the genetically encoded Ca$^{2+}$ indicator GCaMP6s under the pan-neuronal promotor *elavl*3.

The authors first used immuno-histochemistry to confirm that GFP-positive GCaMP-containing cells in the gill filament coincided with serotonergic gill O$_2$ chemoreceptors [i.e. neuroepithelial cells (NECs)], characterized by well-defined markers (synaptic vesicle protein 2 (SV2) and 5-HT). These NECs are polarized towards the distal tips of the filament epithelium, near the efferent filament artery. Conveniently SV2-positive local, intrinsic neurons (ChNs) that formed a chain along the proximal-to-distal aspect of the filament were also GFP positive.

With the aid of a novel *ex vivo*, intact, gill filament preparation the authors monitored hypoxia-evoked changes in intracellular Ca$^{2+}$ levels in NECs or ChNs separately, or in combination. The Ca$^{2+}$ signals in the NECs were robust, stable and readily reversible even after multiple applications of the hypoxic stimulus, before, during and after a variety of drug treatments. Together, these are noteworthy and commendable achievements, and the study stands alone as one that reliably monitored hypoxia-evoked responses in both pre- and postsynaptic sensory cells in their native configuration, using relatively non-invasive methods. This represents a significant advance in a field dominated for the past ~36 years by the application of more invasive techniques to study chemosensory mechanisms in freshly isolated single or cultured cell preparations. The present study convincingly demonstrated that gill NECs function as O$_2$ chemoreceptors when present in their native environment. The hypoxia-evoked Ca$^{2+}$ signal in the NECs depended on both extracellular Ca$^{2+}$ entry through voltage-gated L-type Ca$^{2+}$ channels and Ca$^{2+}$ release from intracellular stores. Moreover this Ca$^{2+}$ signal was inhibited during co-application of exogenous DA and D2R agonists, and was potentiated by D2R antagonists. As commonly observed for D2R signalling, the DA-D2R inhibitory pathway in the NECs appeared to be negatively coupled to adenyl cyclase, reminiscent of mammalian CB O$_2$ chemo-receptors (Nunes et al., 2014). Because D2Rs are preferentially expressed in the NECs, and ChNs exhibit a dopaminergic phenotype, these results led to a model whereby postsynaptic activation of ChNs during hypoxia caused DA release from their nerve endings, leading to negative feedback modulation of NEC responses via D2R signalling.

Several lines of evidence confirmed that ChN responses to hypoxia were indirect or postsynaptic, and dependent upon synaptic transmission from nearby NECs. First during simultaneous paired recordings the hypoxia-evoked Ca$^{2+}$ responses in the ChNs always followed that in the NECs. Second filament transection, aimed to sever neural connections between the more proximal ChNs and distally located NECs, resulted in the loss of hypoxic sensitivity in those remaining ChNs. Third

chemical destruction of ChN dopaminergic nerve terminals, using the neurotoxin 6-hydroxydopamine, similarly eliminated ChN responses to hypoxia. Importantly after both 'denervation' procedures the viability of ChNs was confirmed by their ability to elicit Ca$^{2+}$ responses to high K$^+$ stimulation. Overall these findings strongly support the model that NECs function as presynaptic O$_2$ sensors, whereas ChNs function as postsynaptic sensory afferent neurons.

Although many questions remain unanswered, including the relationship between the ChNs and the extrinsic IXth and Xth cranial nerves that relay chemo-sensory information to the CNS, this groundbreaking study has laid a platform for future exciting studies on the role of branchial O$_2$ chemoreceptors in aquatic vertebrates. Because isolated NECs *in vitro* are capable of sensing other modalities such as high CO$_2$/H$^+$, NH$_3$ and lactate, this intact gill preparation will permit future studies on how these stimuli affect the physiology of NECs and their neural connections in their native environment. The preparation is also attractive for studies on plasticity of NEC-sensory neuron signalling pathways during development and after acclimatization to various stressors present in aquatic environments such as chronic and intermittent hypoxia. Whereas the focus here has been on the inhibitory DA-D2R signalling pathway, the role of excitatory neurotransmitters in gill O$_2$ chemoreception remains to be elucidated. In addition to 5-HT ACh is capable of stimulating cardiorespiratory responses in fish; however cholinergic cells are a separate population, distinct from serotonergic NECs in the branchial epithelium (Pan et al., 2022).

Finally this zebrafish model is expected to provide additional insights into the evolution of O$_2$-sensing mechanisms. Functionally gill NECs exhibit many of the well-studied characteristics of mammalian CB O$_2$ chemoreceptors (i.e. glomus cells). Recent single-cell trans-criptome studies in zebrafish NECs have identified highly expressed genes, including the transcription factor *HIF2α*, and specialized mitochondrial subunit genes, for example, *ndufa4l2a* (Pan et al., 2022), which have been linked to the exquisite acute O$_2$ sensitivity of CB glomus

The Journal of Physiology

cells (Ortega-Sáenz et al., 2020). Due to the advantages of zebrafish as a tractable model for genetic manipulations, exciting opportunities await the use of this whole gill preparation for understanding the origins of vertebrate $O_2$ sensing from an evolutionary perspective.

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

## Additional information

### Competing interests

The author declares no competing interests.

### Author contributions

C.A.N.: conception or design of the work; drafting the work or revising it critically for important intellectual content; final approval of the version to be published; agreement to be accountable for all aspects of the work.

### Funding

None.

### Keywords

calcium imaging, dopamine release, oxygen sensing, zebrafish

### Supporting information

Additional supporting information can be found online in the Supporting Information section at the end of the HTML view of the article. Supporting information files available:

**Peer Review History**

