## [Peer Review History · The Journal of Physiology]

Dopamine inhibits gill O₂ chemoreception: A fishy tale excites

Colin A. Nurse
DOI: 10.1113/JP288732

Corresponding author(s): Colin Nurse (nursec@mcmaster.ca)

The following individual(s) involved in review of this submission have agreed to reveal their identity: Michael G Jonz (Referee #1)

Review Timeline:

Submission Date:	16-Feb-2025
Editorial Decision:	27-Feb-2025
Revision Received:	28-Feb-2025
Accepted:	04-Mar-2025

Senior Editor: Harold Schultz

Reviewing Editor: Andrew Holmes

Transaction Report:

Dear Dr Nurse,

Re: JP-P-2025-288732 "**Dopamine inhibits gill O₂ chemoreception: A fishy tale excites**" by Colin A. Nurse

Thank you for submitting your manuscript to The Journal of Physiology. It has been assessed by a Reviewing Editor and by 1 expert referee and we are pleased to tell you that it is acceptable for publication following satisfactory revision.

The review comments are copied at the end of this email.

Please address all the points raised and incorporate all requested revisions or explain in your Response to Referees why a change has not been made. We hope you will find the comments helpful and that you will be able to return your revised manuscript within 2 weeks. If you require longer than this, please contact journal staff: jp@physoc.org.

REVISION CHECKLIST:

We look forward to receiving your revised submission.

Yours sincerely,

Harold Schultz
Senior Editor
The Journal of Physiology

EDITOR COMMENTS

Reviewing Editor:

Thank you for putting together this excellent perspectives. There are just a couple of minor modifications suggested by the

reviewer that need to be addressed.

Senior Editor:

Thank you for submitting your perspective article to the Journal of Physiology for consideration for the focus article P-RP-2025-287824R3 "Oxygen chemoreceptor inhibition by dopamine D2 receptors in isolated zebrafish gills". The article has been reviewed by the focus authors and found to be acceptable for publication, pending addressing a few minor concerns raised. We should be able to handle a revised manuscript in-house and provide a rapid turnaround. Please submit a revised article addressing the suggestions. Thank you for providing this contribution and support for our Journal.

REFeree COMMENTS

Referee #1:

We thank the author for writing this commentary based on our recent manuscript. The author was very complimentary about our work, and accurately pinpoints the significance of our manuscript. The present commentary will certainly increase the impact of our paper.

I have only very minor comments for the author. They are as follows:

Line 2. Typo in title. Remove "of" to read "Dopamine inhibits gill O2 chemoreception: A fishy tale excites". The title is otherwise correct in the submission system.

Lines 34-35. Add "cells" to read: "(i.e. neuroepithelial cells or NECs)".

Line 59. Suggest use the singular form of ChN: "ChN responses". And also in line 64 ("ChN dopaminergic nerve terminals") and line 65 ("ChN responses").

Line 91. Suggest change from "whole filament gill preparation" to "whole gill preparation" for simplification.

END OF COMMENTS

Response to reviewers

I have only very minor comments for the author. They are as follows:

Line 2. Typo in title. Remove "of" to read "Dopamine inhibits gill O₂ chemoreception: A fishy tale excites". The title is otherwise correct in the submission system.

-THIS ERROR HAS BEEN CORRECTED

Lines 34-35. Add "cells" to read: "(i.e. neuroepithelial cells or NECs)".

-DONE AS REQUESTED

Line 59. Suggest use the singular form of ChN: "ChN responses". And also in line 64 ("ChN dopaminergic nerve terminals") and line 65 ("ChN responses").

DONE AS REQUESTED

Line 91. Suggest change from "whole filament gill preparation" to "whole gill preparation" for simplification.

DONE AS REQUESTED

Dear Professor Nurse,

Re: JP-P-2025-288732R1 "**Dopamine inhibits gill O₂ chemoreception: A fishy tale excites**" by Colin A. Nurse

We are pleased to tell you that your paper has been accepted for publication in The Journal of Physiology.

Yours sincerely,

Harold Schultz
Senior Editor
The Journal of Physiology

If you would like to receive our 'Research Roundup', a monthly newsletter highlighting the cutting-edge research published in The Physiological Society's family of journals (The Journal of Physiology, Experimental Physiology, Physiological Reports, The Journal of Nutritional Physiology, and The Journal of Precision Medicine: Health and Disease), please click this link, fill in your name and email address and select 'Research Roundup':

<https://www.physoc.org/journals-and-media/membernews>

- You can help your research get the attention it deserves! Check out Wiley's free Promotion Guide for best-practice recommendations for promoting your work at: www.wileyauthors.com/eoo/guide. You can learn more about Wiley Editing Services which offers professional video, design, and writing services to create shareable video abstracts, infographics, conference posters, lay summaries, and research news stories for your research at: www.wileyauthors.com/eoo/promotion.

The Corresponding Author will receive an email from Wiley with details on how to register or log-in to Wiley Authors Services where you will be able to place an order

EDITOR COMMENTS

Reviewing Editor:

Thank you very much for making the suggested amendments and for putting together this excellent perspectives article. This will go very well alongside the associated research article and will help to promote this area in general.

Senior Editor:

The editors thank the author for these minor adjustments to a well-written perspective article. The article is now accepted for publication. Congratulations for an interesting and insightful article. Please consider the Journal of Physiology for your future work.